# Approaches to improving patient safety in integrated care: a scoping review

Mirza Lalani [ID] ,[1] Sarah Wytrykowski,[2] Helen Hogan[1]

## ABSTRACT

**Objectives** This scoping review aimed to establish the approaches employed to improving patient safety in integrated care for community-dwelling adults with long-term conditions.

**Design** Scoping review.

**Setting** All care settings.

**Search strategy** Systematic searches of seven academic and grey literature databases for studies published between 2000 and 2021. At the full-text review stage both the first and second reviewer (SW) independently assessed full texts against the eligibility criteria and any discrepancies were discussed.

**Results** Overall, 24 studies were included in the review. Two key priorities for safety across care boundaries for adults with long-term conditions were falls and medication safety. Approaches for these priorities were implemented at different levels of an integrated care system. At the micro-level, approaches involved care primarily in the home setting provided by multi-disciplinary teams. At the meso-level, the focus was on planning and designing approaches at the managerial/organisational level to deliver multi-disciplinary care. At the macro-level, system-wide approaches included integrated care records, training and education and the development of care pathways involving multiple organisations. Across the included studies, evaluation of these approaches was undertaken using a wide range of process and outcome measures to capture patient harm and contributory factors associated with falls and medication safety.

**Conclusions** For integrated care initiatives to fulfil their promise of improving care for adults with long-term conditions, approaches to improve patient safety need to be instituted across the system, at all levels to support the structural and relational aspects of integrated care as well as specific risk-related safety improvements.

[1]Department of Health Services Research and Policy, London School of Hygiene & Tropical Medicine, London, UK
[2]Medical Research Network, London, UK

**Correspondence to**
Dr Mirza Lalani;
mirza.lalani@lshtm.ac.uk

## INTRODUCTION

Health and social care systems globally are facing the challenges of an ageing population with complex comorbidities that require multi-faceted care provision to better manage their health and care needs.[1] Care provision for adults (particularly older and frail) with long-term conditions can be fragmented, impacting the quality of care, with a greater potential for adverse outcomes.[2] The integration of health and social care systems and services is seen as integral to tackling these challenges.[3] The term 'integration' broadly represents a 'joining up' of traditional silos of care across (horizontal) and within (vertical) systems, organisations and services. Integrated care transcends longstanding professional, organisational and regulatory boundaries to provide more coordinated, collaborative and person-centred care.[4–6] Many high-income countries have taken steps to deliver integrated care through integrated care systems which involve the collaboration of sectors and care providers, bringing together primary and secondary healthcare, social care, community and mental health services and voluntary organisations.[7]

The WHO defines patient safety as 'the absence of preventable harm to a patient during the process of healthcare and reduction of risk of unnecessary harm associated with healthcare to an acceptable minimum'.[8] Safer care, involves robust risk management—early identification, continuous monitoring and swift action to address risks.

Integrated care may contribute to improving the safety of care for people with long-term conditions at different levels of the system, particularly by addressing the risks posed by multiple referrals, handovers and discharges that arise as the responsibility for care crosses care boundaries and

---

**STRENGTHS AND LIMITATIONS OF THIS STUDY**

⇒ Highlighting how key components of integrated care can also improve patient safety across care boundaries particularly in reducing falls and medication safety management.

⇒ Another strength of the study was the rigorous approach to scoping and selecting the literature including the comprehensive searching of several academic and grey literature databases and sources.

⇒ Nonetheless, the identification of relevant studies was possibly hindered by the lack of consistency and interchangeability of the terms 'integrated care and patient safety' and their use in the literature.

⇒ While the review was able to identify the different types of patient safety approaches (scope) it was more challenging to describe their scale across care boundaries.

BMJ

involves different practitioners, teams, organisations and systems.[9] Integrated care provides health and care professionals with knowledge of an individual person's diverse care needs through shared information systems, enabling service providers to minimise gaps in treatment and care and address conflicts or duplication in care plans. This creates an environment that supports safety practices including proactive care through timely identification and enhanced management of risk. Furthermore, person-centred care (a key tenet of integrated care) supports people to manage their own care more effectively, ensuring care is tailored to their specific needs and hence, could be regarded as a precursor to safer care as individuals are more aware of the risks of their care activities and their role in mitigating these.[10] With this in mind, strengthening some of the key structural and relational aspects of integrated care[11] such as data sharing, communication, partnership working, care coordination and person-centred care may improve patient safety by enhancing risk management. Patient safety does not necessarily arise from integrated care but from the opportunity for enhanced risk management.

Integrated care initiatives encourage care provision closer to home that is, in the community setting which are safer environments compared with hospitals.[12] However, adverse events in this setting are responsible for between 8% and 12% of hospital admissions, around a half of which are thought to be preventable.[13 14] Harm in this setting may result from medication mismanagement, poor infection control, failure to ensure safe mobilisation or a lack of prevention of and care for pressure sores/ulcers.[15] Studies have also highlighted that the 'spaces' between care settings are prone to problems of communication and coordination which affect the safety of care as patients transition across care boundaries and receive care from different professionals.[16 17]

A handful of conceptual models of integrated care have been described in the literature. One such model suggests integrated care has a supporting role at all levels of the system, micro (services), meso (organisations) and macro (systems).[18] Recent approaches to better management, assurance and improvement of patient safety have advocated for developing safety governance through a whole systems approach addressing safety issues at all levels of a care system.[19]

Initial scoping of the literature on how safety is improved for adults with long-term conditions in integrated care revealed a limited evidence based on the scale and scope of safety approaches implemented in integrated care systems. As the relevant literature to address this issue was likely to be extensive, complex and potentially heterogeneous, a scoping review was selected over a systematic review.[20] This scoping review aims to address this gap by establishing the approaches employed to improve patient safety in integrated care for community-dwelling adults with long-term conditions and how these are evaluated. Such assessments will gauge current key priorities for safety within integrated care and the types of patients safety measures that may be useful for ongoing monitoring of safety at different levels of a system.

## METHODS

This scoping review was conducted in accordance with the Preferred Reporting Items for Systematic Reviews and Meta-Analyses Extension for Scoping Reviews guidance.[21] A protocol was developed but not published. We used Arksey and O'Malley's framework in undertaking this scoping review: (1) identifying the research question; (2) identifying relevant studies; (3) selecting studies; (4) charting the data; (5) collating, summarising and reporting the results.

### Stage 1: identifying the research questions

The scoping review explored the breadth of literature to answer the research question, 'What type of integrated care approaches exist at different system levels to improve patient safety for community-dwelling adults with long-term conditions and how have they been assessed?'.

### Stage 2: identifying relevant studies

Free text search terms were developed based on our research question. The search term strategy (online supplemental file 1) was broadly based on three blocks of terms; (1) population (elderly/vulnerable adults/long-term conditions), (2) approach (associated with integrated care/multi-disciplinary working) and (3) outcome (related to patient safety/risk reduction/preventable harm).

A comprehensive search strategy was employed to include both the peer-reviewed and grey literature. Seven electronic databases were searched including MEDLINE, Social Policy and Practice, Web of Science, CINAHL, Google Scholar, Proquest and OpenGrey. These were supplemented by forward and ancestry citation searches using reference lists of several relevant studies.

### Stage 3: selecting studies

Study selection was based on the inclusion and exclusion criteria listed in table 1.

The first reviewer (ML) undertook initial searches and based on information in the title and abstract selected those meeting criteria for full-text review. At this stage, any studies that the first reviewer was uncertain about in terms of meeting eligibility criteria were checked by the second reviewer. At the full-text review stage both the first and second reviewer (SW) independently assessed full texts against the eligibility criteria and any discrepancies were discussed.

### Stage 4: charting the data

The first reviewer extracted and tabulated the data from the included studies in Microsoft Excel. Details of the studies extracted included the overall aim, population, study design, level of system, integrated care approach, setting and approach to measuring safety. The second reviewer reviewed and verified the data extraction table.

**Table 1** Overview of exclusion and inclusion criteria

| Inclusion criteria | |
|---|---|
| Organisational setting | All types that is, acute, primary, community, mental health and social care |
| Population | Adults over 18 with long-term conditions |
| Approach | Involved a team or group of professionals representing different sectors, for example, acute and social care, often referred to as a multi-disciplinary team (MDT) |
| Outcome | Associated with patient safety as defined by the WHO |
| Study design | Primary and/or secondary data (research studies, quality improvement project reports, service evaluations and reviews) |
| Time interval | 2000–2021: we selected 2000 as a start date because it coincides with a proliferation in safety-related research stimulated by the publication of the seminal report; To Err is Human: Building a Safer Health System.[21] |
| | Studies published in English only |
| **Exclusion criteria** | |
| Organisational setting | Transitions of care within the same sector, for example, hospital only<br>Studies in which it is challenging to discern whether staff in an MDT were employed by different providers and therefore were part of an integrated care team |
| Type of study | We also excluded commentaries and editorials |

Descriptions of the levels of the system, that is, micro, meso and macro and the approach to measuring safety (outcome, process and qualitative)[22] are outlined in box 1. The nomenclature for different levels of a system (micro, meso and macro) can be ascribed to both integrated care and patient safety system governance.[19 23] The Donebedian framework for measuring quality in healthcare applies to identifying measures for safety that is, outcome and process, in the case of this review.[24] We have also added a further measure, 'qualitative' to capture perceptions of study participants regarding service improvements associated with patient safety through implementation of approaches.

---

**Box 1   Description of the different levels of a care system**

**Levels of care system**

Micro-level: typically comprises an approach at the patient level where multiple professionals from different types of organisations provide care commonly in the home setting but sometimes in primary care alone or in primary and secondary care if linked by a common pathway.

Meso level: approaches at organisation level that involve multi-professionals in planning or designing local structures, care pathways and services that span two or more different types of organisations.

Macro level: system wide approaches that may involve several providers with a typical focus on strategy, infrastructure development such as shared care records and other wide-ranging cultural and educational initiatives. Approaches implemented over a wider a geography within a country, for example, region, county, state, etc.

**Approaches to measuring safety**

Outcome measures: indicate the impact of a healthcare service or approach on the patient's health status.

Process measures: practices delivered in order to achieve the desired outcome of an approach.

Qualitative outcomes: perspectives of study participants on the implementation and/or impact of an approach (primarily from qualitative studies only).

---

### Quality assessment

Due to the heterogeneity of the studies in terms of their methodological design, settings in which an approach was implemented as well the design and scope of the approach itself, and in line with similar scoping reviews, formal quality assessment was not undertaken.[25 26]

### Patient and public involvement

No patients or members of the public were directly involved in this work. However, this study is part of a wider programme of research for which there is a research advisory group whose membership includes patients/public representatives. In the conceptualisation phase of the study, the research advisory group provided advice and guidance on the study's aims and purpose.

## RESULTS
### Stage 5: collation, summarising and reporting

The original searches yielded 2957 records. After removing duplicate records and screening of titles and abstracts, 86 were allocated for full-text review. After application of eligibility criteria, 24 studies were included in the review. The flow of studies from initial identification to final inclusion are presented in figure 1.

### General characteristics of included studies

All included studies were published between 2007 and 2019. We identified 1 review[15] and 23 empirical studies. A breakdown of the study characteristics is provided in online supplemental table 1. In terms of geographical distribution, empirical studies were undertaken in North America,[27–32] Europe[33–43 44–46] Asia[47 48] and Australia.[49] All studies involved adult participants described as elderly, older adults or adults over the age of 65 in the main, although four studies also included other adults aged

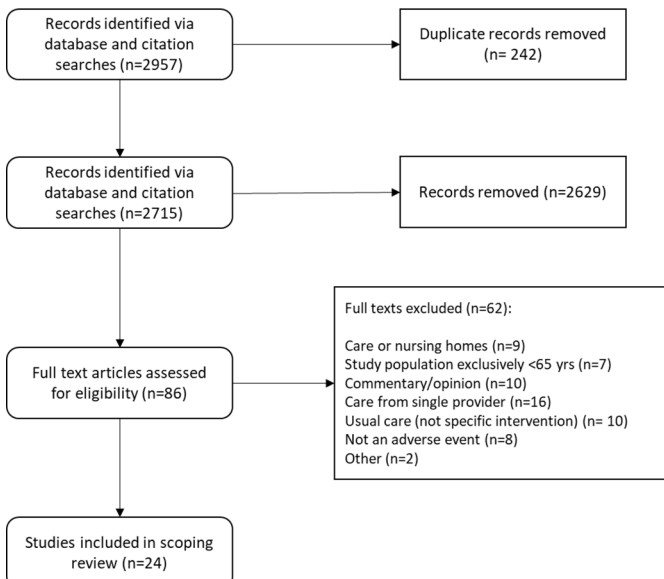

**Figure 1** Flowchart of study selection.

over 18 years. In all studies, the population in receipt of an approach were reported to have long-term, chronic physical or mental health conditions. However, in a few of these studies, a health condition(s) was not specified and instead participants subject to the approach were described as frail or vulnerable. Empirical studies comprised the majority of study types and included randomised control trials,[37 38 40–42 44 49] qualitative studies,[35 36 39 45] mixed methods studies,[27 34] retrospective analysis of medical records,[30 43] a cohort study,[48] a cross-sectional study,[33] an observational study,[36] a systematic review[15] and action research.[46] Service evaluations[28 31 32] and quality improvement (QI) studies[29 47] were also among the list of included studies.

### Types of risks addressed

Of the 24 included studies, 11 involved approaches designed to reduce falls risk,[27 28 32 34 37 38 41 44 46 48 49] and 10 studies were broadly associated with assuring medication safety through reconciliation and review, surveillance and monitoring.[29 33 35 36 39 40 42 43 45 47] Three studies undertook a series of approaches to reduce adverse events more broadly such as identifying medications that increase the risk of a fall in older people and an assessment of an approach designed to reduce the rate of hip fracture.[30 31] The systematic review included studies that involved approaches to reduce adverse events including adverse drug reactions.[15]

### Integrated care approaches
#### Setting

The studies included a range of settings. A total of 13 studies described approaches implemented in a single setting involving a group or team of professionals from different sectors; the home setting[31 34 35 38 41 42 48 49] and the remainder in hospitals,[30 32 36] community health centres[29] and primary care.[45] In the remaining studies,

a care pathway was established either across several settings[15 28 46 47] or just two settings; between the hospital and home setting[37 39 44] or hospital and primary care.[27 33 40 43]

### Level of care system

Various approaches were implemented at different levels of the care system; micro, meso and macro. Ten studies were at the micro-level,[30 34 36–38 41 42 44 48 49] five at the meso-level[15 31 32 40 45] and nine at the macro-level.[27–29 33 35 39 43 46 47] Broadly, these approaches can be categorised into four types: (1) multi-disciplinary team (MDT) approach to care delivery in community (home) or primary care setting, (2) care pathway, (3) educational initiative and (4) integrated digital record (accessible to different care providers).

#### Micro-level

At the micro-level, approaches were characterised by the delivery of person-centred coordinated care for adults with long-term conditions involving the home (community) setting or primary care setting alone or as part of a care pathway with secondary care settings. Care was provided by a range of health and care professionals working together. Three studies outlined multi-disciplinary approaches to prevent and reduce falls, primarily in the home setting such as exercise therapy, home-based assessments for hazards with accompanying modifications, vision assessment, home medication review and a 24-hour call service staffed by a home intervention team.[34 38 49]

A further five studies described approaches implemented across different parts of a care pathway to reduce falls, although Wallace *et al* focused on the development of a multi-disciplinary care pathway following a hip fracture.[30 37 41 44 48] Approaches comprised multi-disciplinary outpatient clinics (often located in hospitals) where patients were assessed and monitored, with further follow-up care provided by community services staff in the home setting.[37 41 44 48] Follow-up care included assistance with daily living, medication review, balance and cognition,[37 44 48] as well as promotion of patient/user independence through home hazard (home environment risk) assessments,[41 44 48] exercise to improve mobility and strength[37 41 48] and patient education on use of assisted devices to support individuals to navigate their home environment.[37]

Three studies involved medicines reviews to reduce the risk of adverse drug events (ADE) including the reduction of falls.[31 42 45]

#### Meso-level

At the meso-level approaches were mainly focused on multi-professional management to plan and design services across at least two different types of organisation, primarily hospitals and community care. Approaches tackled areas with overarching safety risks such as poor communication or dispersed leadership. All five studies used different approaches.

Three studies aimed to improve medication safety for older people. One study employed an Integrated Medicines Management model (spanning hospitals and primary care) which comprised medicines reconciliation and review, patient counselling on medicines usage during hospital stay and greater provision of information and communication with primary care at discharge and postdischarge.[40] Another study described an approach in primary care involving managers, pharmacists and nurses who engaged in self-assessment, peer review, feedback and agreement for change to improve medicines safety management.[45] Willis *et al* described an MDT approach involving medication reviews, home fall risk assessments and blood pressure monitoring.[31]

A further study, outlined a multi-disciplinary falls committee (within a newly created integrated care system) whose role was to provide leadership, direction and coordination to frontline staff promoting new evidence based practices and QI support to prevent falls.[32] The systematic review described approaches such as education and training, organisational/culture interventions (eg, discharge protocol and planning, medication reconciliation, etc) or patient/family orientated approaches concerning awareness and empowerment for patients.[15]

*Macro-level*

At the macro-level, system-wide approaches centred on managing medication safety and reducing falls risk such as key infrastructure developments—integrated care records, training and education initiatives spanning several organisations within a system and creating care pathways involving multiple organisations. Approaches were generally implemented across a wider geography, for example, county, state or region.

Three studies described digital approaches to medication safety monitoring such as a digital database for patients transitioning from secondary to primary care with indicators to alert practitioners to potential ADE[33] and a digital measurement tool designed to assess transitional safety incidents affecting older people including ADE.[43] A further study described how an e-message system for nurses in hospital and community settings could support information exchange, minimising ADE.[39]

Two studies described educational approaches comprising training of inter-professional teams (including social workers) from a range of settings on developing strategies for preventing falls in accordance with national evidence-based guidelines.[27 28]

The remaining studies described multi-component models such as QI collaborative approaches, or other models comprising multi-disciplinary working, care coordination and strategies to strengthen leadership to promote patient safety and reduce hospital admission or readmission. The QI collaborative aimed to reduce the risk of ADE and involved MDTs with professionals from primary and acute care who adopted a change package involving clinical pharmacy services[29] and a QI approach

based on an adaptation of the Institute of Healthcare Improvement Trigger Tool method.[47]

## Profile of professionals involved in the approaches

Of the 23 empirical studies, the majority included a multi-disciplinary approach to care provision involving different health professional groups. Four of the approaches also included social workers as part of the team.[27 28 30 32] In eight studies, the approaches included health and social care professionals such as nurses, paramedics, managers, social workers and allied health professionals; physiotherapists and occupational therapists, pharmacists and optometrists but doctors were not involved.[28 31 32 35–37 39] Many of these studies involved approaches implemented in the community (home setting) or where the home was included as a part of a care pathway.

## Measures for assessing impact of the approach

There was heterogeneity in the approaches to measuring safety. We categorised measures as process, outcome or qualitative.

Two studies included process measures alone. The first involved the monitoring and surveillance of medication safety including the proportion of older patients prescribed falls risk medication, and the proportion of medicines prescribed that have a high potential for major drug interaction.[31] The second study identified measures associated with educational outcomes of health professionals participating in a falls training workshop including subsequent changes to practice. This included workshop participants self-reporting their undertaking of assessments of older patients at higher risk of falls and screening of falls rates.[28]

In 11 studies, outcome measures alone were mentioned. Four of these studies stated falls incidence during follow-up over a period of 6–12 months[37 38 44] as a key measure.[46] In three further studies, falls and falls related injuries such as hip fracture, and hospital length of stay were key outcomes.[30 48 49] In four studies, ADE were stated as outcome measures including preventable ADE or potentially preventable ADE,[33 47] proportion of ADE detected in a specified population[29] and ADEs leading to unplanned hospital admission.[43]

Six studies included both process and outcome measures. Process measures associated with falls included adherence to falls prevention strategies, proportion of falls risk assessments undertaken, documenting of falls and medical history risk of falling, and level of independence in activities of daily living.[27 40 41] Outcome measures in these studies included a decrease in falls rates, time between the first and second falls, falls related subsequent emergency department visits and hospitalisation.

Process measures associated with medication safety comprised medication appropriateness and use of harmful medications.[32 42] Outcome measures in these studies were medication related readmissions, rate of hip fractures and clinically significant drug interactions. The systematic review described a range of process and

outcome measures across the included studies, all associated with adverse events including ADE.[15]

The remaining studies described qualitative outcomes for the integrated approaches i.e. based on the perceptions of study participants. These included assessing approaches designed to prevent falls such as the perceived usefulness of an MDT home-based fall prevention programme.[34] A further four studies described qualitative outcomes in terms of assessing the quality of medication safety management practices. These included the extent of professional learning experiences to support medicines management,[35] the quality of service delivery of a domiciliary hospitalisation unit and its impact on assuring drug safety,[45] the usefulness of a self-assessment tool in identifying areas of improvement for medication safety[36] and the ability of an e-message system to support information exchange to assure medication safety.[39]

## DISCUSSION

This scoping review brings together the published literature on approaches to improve patient safety in integrated care for community-dwelling adults with long-term conditions and demonstrates the range of approaches employed in safety improvement strategies for care across boundaries within several countries. The approach identified in the included studies focus primarily on two common patient safety targets for harm, falls and ADE through falls prevention and medication safety programmes spanning micro, meso and macro levels of a health and care system. Just over a third of the included studies involved the implementation of system-wide approaches (macro-level) such as digital shared care records, training/educational initiatives and the strengthening of care pathways between providers. A higher proportion (40%) of studies described micro-level approaches, primarily MDT delivering care to older people in the home setting alone or as part of a care pathway involving hospitals and primary care. The review also highlighted that the evaluation of approaches indicated a perspective of safety focused primarily on measurement—changes in rates of adverse events or contributory factors at individual patient level.

A key strength of this study was the highlighting of how key components of integrated care can also improve safety across care boundaries particularly in reducing falls and medication safety management. Instituting integrated care practices at the different levels of a care system, for example, digital shared care records (macro), strengthening of care pathways (macro, meso and micro) and promoting MDT working (meso and micro) will strengthen safety approaches and reduce the risk of harm across an integrated care system. The study findings also highlight how in some cases MDT working, particularly in community care does not directly involve doctors. This reflects an increasing trend in some Western European countries where integrated community care teams (without doctors) are primarily responsible for care provision in the home setting for older and/or frail

patients with complex care needs, partly promulgated by the extension of roles of community care professionals and a shortage of primary care doctors.[50 51] The study also identified a range of measures to assess patient safety across systems suggesting that integrated care systems will need to develop a range of indicators to capture the variation in performance of key processes that lead to desired safety outcomes. Another strength of the study was the rigorous approach to scoping and selecting the literature including the comprehensive searching of several academic and grey literature databases and sources. In comparison to the existing literature on this topic, our study builds on previous work on safety in care transitions but extends beyond the spaces and pathways between organisations at the micro-level to consider system-wide (meso and macro) approaches to improving safety.

The study was not without limitations. Articulating a precise definition for patient safety was challenging, as, in several studies, the term was associated with preventing hospitalisation. Hence, we selected the WHO definition which is broad ranging and encompasses the notion of errors of commission and omission. Nonetheless, the identification of relevant studies was hindered by the lack of consistency and interchangeability of the terms 'integrated care and patient safety' and their use in the literature. We may have screened for word choice rather than presence or absence of a similar term or its description. Despite including a range of search terms associated with patient safety some studies may have been missed. Our search strategy did not include social care as a specific search term, although integrated care including care coordination and multi-disciplinary working increasingly involves social care professionals. We also did not search for 'person-centred care' or 'self-care' both of which are potentially effective risk management strategies. This may explain why we were unable to identify any studies reporting the experience of patients/carers. The nature of our search did not lead to papers that talked about implementation of structures or their measurement and hence we only included process and outcome measures. While the review was able to identify the different types of patient safety approaches (scope) it was more challenging to describe their scale across care boundaries.

Integrated care initiatives identified in this study involved the design of a range of new structures and processes to support the delivery of safer care, although further work is required to establish their effectiveness in improving patient safety. Integrated digital care records have been cited as a key enabler for structural integration in integrated care systems as they promote sharing of information thus minimising the risk of care-related information being overlooked as people (particularly older and more frail) move along a care pathway.[52] Training and educational initiatives delivered to inter-professional teams are also considered to be a means of improving the quality of integrated care delivery, supporting the development of social and working relationships thereby promoting partnership working.[53] At all levels, several approaches

were implemented to strengthen the care pathway on discharge from hospital and provide tailored care in the home setting to reduce the risk of adverse events. These initiatives had several goals associated with integration, to deliver care closer to home, reduce duplication, minimise the risk of hospital admission and provide personalised care.[3] Each of these goals are notable processes of integration which, if successfully implemented, could also improve patient safety outcomes. These focused approaches do not negate the need for further efforts to develop patient safety within integrated care systems including safety governance at all levels, system-wide mechanisms to identify, review and learn from safety incidents, enhanced capacity and capability for measurement and improvement, and creating learning cultures that promote openness and transparency.[19 54]

The levels of the system (micro, meso and macro) approach to identifying integrated care and patient safety approaches presented in this study provides a useful means of understanding how organisations can implement change, improvement and innovation across an integrated care system. At the macro-level, the implementation of integrated care records to collate information and support communication alongside the promotion of a learning culture supported by inter-professional training and education is required. At the meso-level, the focus would be on organisations jointly planning and designing multi-disciplinary approaches to develop seamless care pathways, minimising the risk of harm as people transition between providers. At the micro-level, a range of multi-disciplinary working practices to improve safety could be deployed from near real time safety monitoring within 'community huddles' to protected time sessions to compare and reflect on performance across teams promoting longer-term changes.[55] These approaches are potential solutions to the fragmentation of health and care systems (which continues despite efforts to integrate care) and will be integral to improving quality and safety outcomes for increasing populations of people with long-term conditions as they receive care closer to home.[19]

The studies included in this review evaluated falls reduction and medicines safety approaches using clinical process and outcome measures with several authors looking at associations between these outcomes and wider health system outcomes, for example, hospital (re) admissions and length of stay. System outcome measures have been widely used as performance indicators to assess the impact of integrated care.[3 56] While structural or relational measures of integration such as collaborative and coterminus working[50 57 58] or the aforementioned system outcomes dominate the discourse of the effects of integrated care, less attention has been paid to measures related to patient or carer experience.[59] None of the studies in this review collected data on the experience of patients and carers alongside clinical outcomes. The understanding of experiences of patients in receipt of services may support service improvement to deliver better quality care.[60 61]

A wide range of process and outcome measures were used across the studies predominantly to capture patient harm and contributory factors. Expanding measurement to incorporate the capture of the variation in performance of key processes that lead to desired outcomes, would not only identify the most effective approaches for improving patient safety but also open up a wider range of options to support building resilience across and integrated care system.[62] A standardised set of safety indicators would support comparisons between different integrated care approaches to safety.

Safety approaches in this review were mainly focused on medication safety management and falls prevention which is unsurprising as both account for a notable proportion of unplanned hospital admissions. In England, in 2018 falls accounted for a total of 220 160 hospital admissions in those aged over 65 years. A study of medication errors in the UK estimated that ADE accounted for 27 362 hospital admissions annually, relating to 136 811 bed days at a cost of £83.7 million to the National Health Service.[63 64] While falls and medication safety are justifiably key safety priorities for integrated care, there is scope to broaden the focus to other common safety issues. For example, a study in the Netherlands found that of 106 admissions to hospital in patients aged over 65 over a period of 12 months were due to an adverse event with 45% due to procedural related infections, community acquired infection such as pneumonia with/without signs of sepsis and diabetes related hypoglycaemia.[64 65]

Future research should focus on the role of other patient safety targets (not just falls and medication related safety) such as healthcare associated infections, community acquired infections and pressure ulcers/sores which also increase the risk of harm for people with long-term conditions especially in the community setting. Furthermore, research must establish the effectiveness of the structural and relational aspects of integrated care such as care coordination as well as the approaches identified in this review, for example, integrated care records and education and training, in terms of their impact on reducing the risk of harm across care pathways. This would provide the evidence for policymakers and system leaders to implement the necessary changes to support integration, safety governance and service improvement.

## CONCLUSION

In order to maximise the potential for to fulfil their promise of improving care for people with long-term conditions, approaches such as integrated care records, allocating resources for training and education, greater efforts to form robust and risk-limited care pathways as well as support for MDT working, especially in the community setting will have to be key priorities for improving safety across systems. They also align to the goals of by reducing fragmentation and enhance personalisation of care. Such a focus would ensure that integrated care reaches its full potential as an enabler of safety supporting risk

management and personalisation of care for people with long-term conditions.

**Contributors** ML conceived and designed the study. All authors contributed to several aspects of the review process, that is, stages 1–5 described in the manuscript under the methods section. ML and HH identified the research question. ML developed the search strategy (with support from HH) and identified relevant studies. ML and SW were involved study selection, charting of the data from included studies and results collation, reporting and summarising of the data. ML drafted the manuscript. All authors approved the final version of the manuscript. ML accepts full responsibility for the work and/or the conduct of the study, had access to the data, and controlled the decision to publish.

**Funding** This research is funded by the National Institute for Health Research (NIHR) Policy Research Programme, conducted through the Quality, Safety and Outcomes Policy Research Unit, PR-PRU-1217-20702. The views expressed are those of the author(s) and not necessarily those of the NIHR or the Department of Health and Social Care.

**Competing interests** None declared.

**Patient and public involvement** Patients and/or the public were not involved in the design, or conduct, or reporting, or dissemination plans of this research.

**Patient consent for publication** Not applicable.

**Ethics approval** Not applicable.

**Provenance and peer review** Not commissioned; externally peer reviewed.

**Data availability statement** All data relevant to the study are included in the article or uploaded as supplementary information.

**ORCID iD**
Mirza Lalani http://orcid.org/0000-0001-7851-9062

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
