## [Reviewer comments · BMJ Open]

ARTICLE DETAILS

TITLE (PROVISIONAL)	Approaches to improving patient safety in integrated care: a scoping review
AUTHORS	Lalani, Mirza; Wytrykowski, Sarah; Hogan, Helen

VERSION 1 – REVIEW

REVIEWER	Khalil, Hanan La Trobe University, Department of Rural and Indigenous Health
REVIEW RETURNED	11-Oct-2022

GENERAL COMMENTS	Thank you for the opportunity to review the manuscript. Overall, this is a well written manuscript, however, I have the following comments for the authors to consider. 1. There are inconsistencies in the alignment of the research question, the results and the conclusion. The focus of the scoping review is to map the approaches employed to manage patient safety in integrated care for community dwelling people with long term conditions. The conclusion of the abstract includes evaluation, the conclusion of the manuscript introduces new information and does not summarise the results of the review.2. The concept of the review should be "integrated care approaches for chronic conditions. The authors only rereferred to two or more interventions (please provide a reference for this). these two phrases are not similar.3. Please provide a definition for "integrated care approaches in the introduction.4. I am not sure if the focus is patient safety or system safety, please use consistent terms5. scoping reviews protocols should be published in open science framework. Please refer to this paper Peters, M.D., Marnie, C., Tricco, A.C., Pollock, D., Munn, Z., Alexander, L., McInerney, P., Godfrey, C.M. and Khalil, H., 2020. Updated methodological guidance for the conduct of scoping reviews. JBI evidence synthesis, 18(10), pp.2119-2126.6. Please try and present some of the data in graphical or tabular format. It is hard to include all the results in one table.7. Please provide other examples of patient safety targets in your discussion.
---

REVIEWER	Russell, Grant Monash University, School of Primary Health Care
REVIEW RETURNED	16-Oct-2022

GENERAL COMMENTS	Thank you for the opportunity to review the manuscript: "Approaches to improving patient safety in integrated care: a scoping review" for BMJ Open. The article used a scoping review
---

	approach to establish the approaches employed to manage patient safety in integrated care for community dwelling people with long term conditions. This was a solidly designed piece of work that followed procedures of the increasingly popular scoping review methodology. Apart from some minor editorial points there were a couple of fundamental issues that need to be addressed in any review of the manuscript. There are three areas of concern that I feel would be worth close consideration. The theory and the logic of the study The authors have set themselves a difficult task in this work. The concepts of integration and safety are both complex, however I am concerned that there has been very little use of theoretical considerations to inform the review. – I feel that the review would have benefited considerably to have incorporated a broader conceptualisation of integration – it would have been helpful to refer to Valentijn, P. P., et al. (2013). "Understanding integrated care: a comprehensive conceptual framework based on the integrative functions of primary care." Int J Integr Care 13: e010. In terms of safety, again there seems no real conceptual model – again it would have been worth at least acknowledging models such as the Manchester Patient Safety Framework would have been helpful. The population of focus. Neither the title, the research question nor the abstract characterises a key limitation of the review – the inclusion only of patient populations of individuals aged 65 and above. As with the intervention / approaches issue, the reader of the article would be expecting not to see the restriction. It would be especially irritating for clinicians and consumers aware of the challenges faced by younger individuals living with long term disabilities, especially those requiring long term integrated care. A blurring of approaches and interventions. The title and the stated objective of the scoping review is to identify approaches that have been used to manage patient safety in integrated care. By contrast, the review seems devoted to identifying interventions. The confusion is compounded by the fact that the background speaks of the varied levels of systems of care, and of the structures that have been used to address and incorporate safety. Then, at line 79 on page 4 the language transitions to speaking of interventions. The remainder of the article then speaks about interventions. Consistency would help the reader. Otherwise I have made a few comments about the methods and some editorial issues Methods The paper seems to follow the Arksey and O'Malley approach (although I would suggest that the authors reword the phrase 'We used the first five stages of Arksey and O'Malley's framework...' (p4 line 89) – it gives the impression that a phase was left out. The 'left out phase is optional in the framework. As far as the methods were concerned, there is a phrase that justified why a scoping review is the most appropriate review in the methods (p5 line 91) – it would be better in the background as the justification. It is not entirely clear as to how the authors have a) identified the gap in understanding and b) decided that a scoping review is the way to fill the gap. Or, as is said in PRISMA-Scr "Describe the rationale for the review in the context of what is
--	--

	already known. Explain why the review questions/objectives lend themselves to a scoping review approach” Finally, I was surprised that only one reviewer screened titles and abstracts – I couldn't be clear as to how 'borderline' articles were considered by the research team. Editorial issues. Acronyms can occasionally be helpful in improving flow of a document. However the authors have used several acronyms that, if anything make it harder to understand the article. The general reader would be familiar with 'QI' for example, but would have difficulty with LTC ICS ADE, and MDT – they make clarity more difficult. The use of LTC is even more problematic given the acronym usually refers to long term care (as opposed to long term conditions) The written English is otherwise clear, although there are several minor typographical errors through the manuscript (mainly missing prepositions).
--	--

REVIEWER	Berntsen, Gro Norwegian centre for E-health Research, Integrated care
REVIEW RETURNED	22-Oct-2022

GENERAL COMMENTS	General summary Integrated care is thought to promote safer care, through improved coordination and quality of care. This is a scoping review, seeking to tease out the safety process and outcomes measures that arise as a benefit from integrated care interventions. They seek to understand what integrated care interventions that also report on patient safety look like (context, and content), which RQ they answer, and especially, which patient safety issues are addressed, including safety outcomes. The authors have performed a broad search, including the grey literature. They present the identification of relevant studies in a clear and comprehensive manner, including a flow-chart. The authors have applied clear inclusion/ exclusion criteria, They use clear definitions of integrated care (across at least 2 organizations) and patient safety as defined by WHO. They have structured their review by micro- meso and macro-level interventions. The authors included 24 studies. They find that integrated care and safety interventions are mostly focused on medication review and falls preventions. The interventions vary widely, from macro: system wide implementations of integrated care EHRs, meso: development of care pathways, to micro: person centered care plans. They conclude that there is a lack of focus on safety specifically in integrated care interventions. I congratulate the authors with a big task, which surely has taken a lot of time and resources. Thank you. The study is methodologically stringent, and the analysis is clear, the language is crisp and clear. Major points Unclear research question. Are you studying the safety effects of integrated care, or are you studying risk management strategies that are often employed together with integrated care approaches?
---

	I agree with the authors, that integrated care could impact safety, as continuity of care, could secondarily impact safety. The authors describe how fragmented care carries an inherent risks for medication errors, pressure sores, infections, unsafe mobilization etc. The authors offer the following theoretical understanding of why integrated care should be safer: "Risks to patient safety can be mitigated within ICS by strengthening some of the key structural and relational aspects of service provision such as communication, partnership working, care coordination and person-centred care as well as intervening to reduce specific harms." In my opinion, person-centered care, and specific risk reduction strategies are not in and of themselves inherent to or part of integrated care. The challenge is to differentiate between risk-management that arises from integrated care, and risk-management approaches that are independent of, and therefore not really a result of integrated care. For instance, a falls prevention program can be done by a physiotherapist alone, or it could be performed by a physio and the pharmacist together. When they combine their competencies, they will create a more comprehensive falls prevention plan than each could do on their own. The latter would be a patient safety improvement which can be attributed to integrated care, whereas the former is not attributable to integrated care. The former would be a risk management strategy, made in a stand-alone context, by the physiotherapist. Another way to think about integrated care and safety is that information always precedes action. The integrated care team may be safer because the care team is sharing information and competencies. Therefore, they can understand and manage risks better, when they have a broader overview of the case, than each of the agents can, when they work on their own. To summarize, I believe it is essential that we learn to distinguish between different strategies of care quality improvement. To reach the quadruple aim, we need to pursue person-centered, integrated and pro-active care or patient safety, simultaneously {Berntsen, 2019 #12330}. We need to understand the essence of each term, and not lump them all together under one heading. Thus the paper would be strengthened by a theoretical reflection on how the authors would expect integrated care to improve safety. Secondly to differentiate between patient safety improvements that arise from integrated care, and those that arise from improved risk-management, but are not really dependent on the integrated care component. I would invite the authors to re-think their research question in terms of these issues. Search strategy The terminology used for patient safety in the search strategy is quite narrow. I miss a comment on the plethora of terms other than patient safety, that could have been part of the search strategy. These are
--	--

	terms which also indicate an active management of risks, such as pro-active care, anticipatory care, risk management, remote follow up, early intervention, self-mangement of risk, etc. Single organization contexts – can they be integrated? In Line167. Please explain how papers set in single organization interventions, such as either hospital or community health centers, can call themselves integrated? Why are doctors not part of integrated care interventions? In line 243: "... but doctors were not involved.." this is very surprising. Were there no doctors involved in any of the integrated care approaches? If this is so, I believe it warrants some reflection in the discussion. Why are doctors not part of the integrated care effort? What role should patients have in patient safety efforts? Line358: " None of the studies in this review collected data on the experience of patients and carers alongside clinical outcomes" I agree that this is an important point. Could this be the result of the way you conducted your search?, i.e. you searched for integrated care, not person-centered care, and thus inadvertently excluded studies that were focused on both topics? If yes – add a sentence on this in weaknesses and limitation. Minor comments: Page 5, line 70: "... to ensure safe mobilisation or a lack of prevention of and care for pressure areas. 12" What kind of pressure are we talking about here? Pressure sores? Or pressure in work for professionals? Acronyms must be spelled out. MDT, ICS, ADE – please provide a full-name for each, before you apply the abbreviation.
--	--

VERSION 1 – AUTHOR RESPONSE

General summary

Integrated care is thought to promote safer care, through improved coordination and quality of care. This is a scoping review, seeking to tease out the safety process and outcomes measures that arise as a benefit from integrated care interventions. They seek to understand what integrated care interventions that also report on patient safety look like (context, and content), which RQ they answer, and especially, which patient safety issues are addressed, including safety outcomes.

The authors have performed a broad search, including the grey literature. They present the identification of relevant studies in a clear and comprehensive manner, including a flow-chart. The authors have applied clear inclusion/ exclusion criteria, They use clear definitions of integrated care (across at least 2 organizations) and patient safety as defined by WHO. They have structured their review by micro- meso and macro-level interventions.

The authors included 24 studies. They find that integrated care and safety interventions are mostly focused on medication review and falls preventions. The interventions vary widely, from macro: system-wide implementations of integrated care EHRs, meso: development of care pathways, to

micro: person-centered care plans. They conclude that there is a lack of focus on safety specifically in integrated care interventions.

I congratulate the authors with a big task, which surely has taken a lot of time and resources. Thank you. The study is methodologically stringent, and the analysis is clear, the language is crisp and clear.

Major points

Unclear research question.

Are you studying the safety effects of integrated care, or are you studying risk management strategies that are often employed together with integrated care approaches?

I agree with the authors, that integrated care could impact safety, as continuity of care, could secondarily impact safety. The authors describe how fragmented care carries an inherent risks for medication errors, pressure sores, infections, unsafe mobilization etc. The authors offer the following theoretical understanding of why integrated care should be safer:

“Risks to patient safety can be mitigated within ICS by strengthening some of the key structural and relational aspects of service provision such as communication, partnership working, care coordination and person-centred care as well as intervening to reduce specific harms.”

In my opinion, person-centered care, and specific risk reduction strategies are not in and of themselves inherent to or part of integrated care. The challenge is to differentiate between risk-management that arises from integrated care, and risk-management approaches that are independent of, and therefore not really a result of integrated care. For instance, a falls prevention program can be done by a physiotherapist alone, or it could be performed by a physio and the pharmacist together. When they combine their competencies, they will create a more comprehensive falls prevention plan than each could do on their own. The latter would be a patient safety improvement which can be attributed to integrated care, whereas the former is not attributable to integrated care. The former would be a risk-management strategy, made in a stand-alone context, by the physiotherapist.

Another way to think about integrated care and safety is that information always precedes action. The integrated care team may be safer because the care team is sharing information and competencies. Therefore, they can understand and manage risks better, when they have a broader overview of the case, than each of the agents can, when they work on their own.

To summarize, I believe it is essential that we learn to distinguish between different strategies of care quality improvement. To reach the quadruple aim, we need to pursue person-centered, integrated and pro-active care or patient safety, simultaneously {Berntsen, 2019 #12330}. We need to understand the essence of each term, and not lump them all together under one heading. Thus the paper would be strengthened by a theoretical reflection on how the authors would expect integrated care to improve safety. Secondly to differentiate between patient safety improvements that arise from integrated care, and those that arise from improved risk-management, but are not really dependent on the integrated care component. I would invite the authors to re-think their research question in terms of these issues.

The reviewer provides some very useful insights. That said, as the aim of the review mentions our primary interest was to understand how integrated care approaches could improve patient safety. We refer to this throughout, including in the discussion sections when we discuss how integrated care at the different levels of the system may improve patient safety by minimising risk. We agree that risk reduction strategies are not inherently part of integrated care approaches – but our study highlights how integrated care could reduce risk across a system. We also believe the person-centred approaches are now a key underlying principle of integrated care approaches, although whether this is adequately delivered upon is an entirely different matter, outside the scope of this study.

Search strategy

The terminology used for patient safety in the search strategy is quite narrow. I miss a comment on the plethora of terms other than patient safety, that could have been part of the search strategy. These are terms which also indicate an active management of risks, such as pro-active care, anticipatory care, risk-management, remote follow up, early intervention, self-mangement of risk, etc.

While we recognise that there was scope to expand the list of terms associated with patient safety, in our consultation with the wider research group (including safety experts) the chosen terms were deemed to adequately align with the definition of patient safety.

Single organization contexts – can they be integrated?

In Line167. Please explain how papers set in single organization interventions, such as either hospital or community health centers, can call themselves integrated?

This has been clarified– line 194-196. Despite the studies being set in a single setting, the interventions delivered involved different professionals across different sectors i.e. acute, primary, social care etc.

Why are doctors not part of integrated care interventions?

In line 243: “... but doctors were not involved..” this is very surprising. Were there no doctors involved in any of the integrated care approaches? If this is so, I believe it warrants some reflection in the discussion. Why are doctors not part of the integrated care effort?

Thank you for highlighting this issue. We have included some brief reflections in the discussion see lines 333-338.

What role should patients have in patient safety efforts?

Line358: “ None of the studies in this review collected data on the experience of patients and carers alongside clinical outcomes” I agree that this is an important point. Could this be the result of the way you conducted your search?, i.e. you searched for integrated care, not person-centered care, and thus inadvertently excluded studies that were focused on both topics? If yes – add a sentence on this in weaknesses and limitation.

Thank you – this has been addressed, lines 355-358.

Minor comments:

Page 5, line 70: "... to ensure safe mobilisation or a lack of prevention of and care for pressure areas. 12"

What kind of pressure are we talking about here? Pressure sores? Or pressure in work for professionals?

Acronyms must be spelled out. MDT, ICS, ADE – please provide a full-name for each, before you apply the abbreviation.

Minor comments have been addressed as advised.

Thank you for the opportunity to review the manuscript: "Approaches to improving patient safety in integrated care: a scoping review" for BMJ Open. The article used a scoping review approach to establish the approaches employed to manage patient safety in integrated care for community dwelling people with long term conditions.

This was a solidly designed piece of work that followed procedures of the increasingly popular scoping review methodology. Apart from some minor editorial points there were a couple of fundamental issues that need to be addressed in any review of the manuscript.

There are three areas of concern that I feel would be worth close consideration.

The theory and the logic of the study

The authors have set themselves a difficult task in this work. The concepts of integration and safety are both complex, however I am concerned that there has been very little use of theoretical considerations to inform the review. – I feel that the review would have benefited considerably to have incorporated a broader conceptualisation of integration – it would have been helpful to refer to Valentijn, P. P., et al. (2013). "Understanding integrated care: a comprehensive conceptual framework based on the integrative functions of primary care." *Int J Integr Care* 13: e010. In terms of safety, again there seems no real conceptual model – again it would have been worth at least acknowledging models such as the Manchester Patient Safety Framework would have been helpful.

We thank the reviewer for their comments. We have addressed this issue by summarising the relevant aspects of Valentijn et al's model and added further detail from the OECD safety governance framework. See table 1.

The population of focus.

Neither the title, the research question nor the abstract characterises a key limitation of the review – the inclusion only of patient populations of individuals aged 65 and above. As with the intervention / approaches issue, the reader of the article would be expecting not to see the restriction. It would be especially irritating for clinicians and consumers aware of the challenges faced by younger individuals living with long term disabilities, especially those requiring long term integrated care.

We grateful to the reviewer for highlighting this issue. The review does in fact include four studies among the included list of studies in which the population of adults is over 18. We have therefore amended the inclusion criteria accordingly. The search terms include 'vulnerable adults' which would have accounted for those adults with long term conditions that are not necessarily over 65. We believe the framing of the inclusion criteria as restricting to over 65 was an oversight on our part.

A blurring of approaches and interventions.

The title and the stated objective of the scoping review is to identify approaches that have been used to manage patient safety in integrated care. By contrast, the review seems devoted to identifying interventions. The confusion is compounded by the fact that the background speaks of the varied levels of systems of care, and of the structures that have been used to address and incorporate safety. Then, at line 79 on page 4 the language transitions to speaking of interventions. The remainder of the article then speaks about interventions. Consistency would help the reader.

Otherwise I have made a few comments about the methods and some editorial issues

We thank the reviewer for pointing this out. We have used approaches throughout to reduce any ambiguity.

Methods

The paper seems to follow the Arksey and O'Malley approach (although I would suggest that the authors reword the phrase 'We used the first five stages of Arksey and O'Malley's framework...' (p4 line 89) – it gives the impression that a phase was left out. The 'left out phase is optional in the framework.

This has been rectified, thank you.

As far as the methods were concerned, there is a phrase that justified why a scoping review is the most appropriate review in the methods (p5 line 91) – it would be better in the background as the justification. It is not entirely clear as to how the authors have a) identified the gap in understanding and b) decided that a scoping review is the way to fill the gap. Or, as is said in PRISMA-Scr “Describe the rationale for the review in the context of what is already known. Explain why the review questions/objectives lend themselves to a scoping review approach”

We have heeded the advice of the reviewer moving the justification sentence to the background section and adding more clarity to the rationale for the study. See lines 87-95.

Finally, I was surprised that only one reviewer screened titles and abstracts – I couldn't be clear as to how ‘borderline’ articles were considered by the research team.

Many thanks for this comment – all studies screened by the first reviewer for which there was uncertainty about eligibility criteria were in fact subject to a further check by the second reviewer. This has been stated in line 132.

Editorial issues.

Acronyms can occasionally be helpful in improving flow of a document. However the authors have used several acronyms that, if anything make it harder to understand the article. The general reader would be familiar with ‘QI’ for example, but would have difficulty with LTC ICS ADE, and MDT – they make clarity more difficult. The use of LTC is even more problematic given the acronym usually refers to long term care (as opposed to long term conditions)

The written English is otherwise clear, although there are several minor typographical errors through the manuscript (mainly missing prepositions).

Acronyms have been addressed as follows:

LTC and ICS has been spelled fully in all cases. We regard ADE and MDT as fairly commonly used and understood acronyms both by practitioners and in the literature.

Thank you for the opportunity to review the manuscript. Overall, this is a well written manuscript, however, I have the following comments for the authors to consider.

1. There are inconsistencies in the alignment of the research question, the results and the conclusion. The focus of the scoping review is to map the approaches employed to manage patient

safety in integrated care for community dwelling people with long term conditions. The conclusion of the abstract includes evaluation, the conclusion of the manuscript introduces new information and does not summarise the results of the review.

We have removed reference to evaluation in the conclusion of the abstract. The conclusion of the paper now aligns with the abstract but also summarises the paper's main findings i.e. the various approaches associated with integrated care that improve patient safety.

2. The concept of the review should be "integrated care approaches for chronic conditions. The authors only rereferred to two or more interventions (please provide a reference for this). these two phrases are not similar.

We thank the reviewer for this comment. To ensure consistency we have removed reference to intervention and have used approaches throughout.

3. Please provide a definition for "integrated care approaches in the introduction.

We have expanded the definition of integrated care in the introduction including adding further relevant content in table 1.

4. I am not sure if the focus is patient safety or system safety, please use consistent terms.

We now refer largely to patient safety other than when discussing outcome indicators for safety systems.

5. scoping reviews protocols should be published in open science framework. Please refer to this paper Peters, M.D., Marnie, C., Tricco, A.C., Pollock, D., Munn, Z., Alexander, L., McInerney, P., Godfrey, C.M. and Khalil, H., 2020. Updated methodological guidance for the conduct of scoping reviews. JBI evidence synthesis, 18(10), pp.2119-2126.

We thank the reviewer for this comment but there is not mandatory requirement to publish a scoping review protocol and we cannot see the value in doing so retrospectively?

6. Please try and present some of the data in graphical or tabular format. It is hard to include all the results in one table. We did consider splitting up parts of the overall table but found that having all the results in one table was the most cogent approach to presenting the results.

7. Please provide other examples of patient safety targets in your discussion.

We thank the reviewer for this comment and have added three further examples of patient safety targets. See line 421-423.

VERSION 2 – REVIEW

REVIEWER	Berntsen, Gro
----------	---------------

	Norwegian centre for E-health Research, Integrated care
REVIEW RETURNED	13-Feb-2023

GENERAL COMMENTS	Berntsen – re-review – 13th of Febr 23 Thank you for the opportunity to re-review this paper. I can see that the authors have tried to address the issues that I commented on in my first review. Major comments: Integrated care and safe care are two conceptually distinct entities: In the literature of managing vulnerable elderly, almost all papers look at a mix of person-centered, integrated and pro-active or patient safety approaches {Berntsen, 2019 #12330}. The intuitive understanding is that these three concepts all need to be addressed together to create quality of care. While I realize that it is difficult to “disentangle” these three concepts, I also believe that if we lump all good things together in to one big safe-care mix, the concepts become unclear and not very useful. The paper reflects a point of view where integrated care seems to lead to safer care, through integration itself. Is this true? I believe Integrated care will promote safer care under certain conditions. It is therefore necessary to clearly distinguish between concepts of person-centered, integrated and safe care, because these terms are synergistic and interdependent. It is also important to describe how integrated care promotes safer care. Care integration is necessary for persons with complex needs when many providers need to collaborate to make sure that management is coordinated. Integrated care in and of itself does not improve care processes or care quality. It is only when lack of knowledge or overview of a case, that there is a risk for gaps, interactions, and conflicts between multiple single disease care plans. To integrate care, allows the agents to become aware of new risks. Risks they are unaware of when they are blind to each others activities. Integrated care is a requirement for pro-active care. When care is integrated, the new risks that are identified can be managed, so that risks can be eliminated or lowered. Safe care or pro-active care is the active identification of risks and management of those risks. It is useful for any patient in an unstable condition, where deterioration might be heralded by subtle changes that are hard to detect. Risk management arises from actively recognizing risks, monitoring risks, and responding to them with action. Person-centered care is the co-creation of a care plan, together with the patient. It is a sharing of power, so that what is known to be effective and meaningful by the patient, is reflected in the care plan. PCC is also a pre-requisite for safe care. Unless the patient finds the risk-mitigation activities both meaningful and feasible, he/she will not implement them. Patients can veto anything we suggest. Therefore, involving and engaging patients are part and parcel of safe care.
---

	I would invite the authors to re-review their introduction in light of a conceptual distinction between integrated care and safe care, while recognizing their interdependencies. Safety does not arise from integration itself. It arises from the improved opportunity for risk management. Table 1 is rather confusing. It seems to me that table 1 descriptions of integrated care conflicts with the excellent descriptions given in Box 1. Also, the patient safety descriptions in Table 1 are not very meaningful to illustrate how care becomes safer as a result of care integration. I suggest that integrated care by level of system is described either in introduction or in methods, and that you retain the descriptions in Box 1. I suggest you consider the following: Safe care arises from integrated care at micro-level:" ... by addressing the risks posed by multiple referrals, handovers and discharges that arise as the responsibility for care crosses care boundaries and involves different practitioners, teams, organisations and systems" AND identifying concrete risks which are then managed by the providers working together. I believe both macro and meso-level integration of services lead to safer care, mainly by supporting a better micro-level opportunity to identify and manage risks. Hence I think table 1 is superfluous. Safety in the community setting: The authors write: "Even so, integrated care initiatives themselves pose a risk to patient safety as care shifts away from hospitals and closer to home. Adverse events in the community setting cause between 8-12% of hospital admissions, around a half of which are thought to be preventable. Harm in this setting may result from medication mismanagement, poor infection control, failure to ensure safe mobilisation or a lack of prevention of and care for pressure sores/ulcers." To me, this paragraph tells me that patients with complex care needs are safer in hospital than at home. However, we know that hospitalization increases the risk of hospital acquired infections and deconditioning. {Smith, 2020 #14379} Being hospitalized is also not a good quality of life. Please revise. Search terms: The search terms are outlined at a very high level in the methods section. Please outline the key conceptual words that would define the areas that you have combined in your search. Single or multiple settings? In methods you state that "inclusion criteria: Involved at least two different types of organisations e.g. hospital". Yet in results you report on several papers set in a "single setting". Then again in results, you explain what you mean with the single setting: "Studies based in a single setting included approaches that involved professionals from different sectors across social, acute, primary and community working
--	---

	together (often in the same team) to provide care.” This is confusing. Please revise and use a term that is intuitive to describe a single team, which consists of professionals representing many settings. Maybe Multidisciplinary team ? Measures for assessing impact: The authors have chosen, appropriately, to use Donabedians Structure > Process > Outcomes to organize their findings. However you report only on process and outcomes measures of impact. Why did you leave out structural measures of impact? Example. The implementation of a policy to increase the number of multi-disciplinary teams (MDT), would be expected to increase the number of MDTs (structures) that can handle integrated care. Self-care and risk management One of the most important ways of both monitoring and reducing risk is to involve and engage patients in this activity. Patients are the only persons present at all times in their own patient journey. They are the ones who have most to gain by effective risk management. They are also much cheaper labor and are always present. It is almost never possible to promote effective risk management without a proper involvement of patients. As the authors did not include self-care as a risk-management focus in their search, I suggest that this is mentioned in the limitations of the study.
--	---

VERSION 2 – AUTHOR RESPONSE

Reviewer 3

Integrated care and safe care are two conceptually distinct entities:

In the literature of managing vulnerable elderly, almost all papers look at a mix of person-centered, integrated and pro-active or patient safety approaches {Berntsen, 2019 #12330}. The intuitive understanding is that these three concepts all need to be addressed together to create quality of care. While I realize that it is difficult to “disentangle” these three concepts, I also believe that if we lump all good things together in to one big safe-care mix, the concepts become unclear and not very useful.

The paper reflects a point of view where integrated care seems to lead to safer care, through integration itself. Is this true? I believe Integrated care will promote safer care under certain conditions.

It is therefore necessary to clearly distinguish between concepts of person-centered, integrated and safe care, because these terms are synergistic and interdependent. It is also important to describe how integrated care promotes safer care.

Care integration is necessary for persons with complex needs when many providers need to collaborate to make sure that management is coordinated. Integrated care in and of itself does not improve care processes or care quality. It is only when lack of knowledge or overview of a case, that there is a risk for gaps, interactions, and conflicts between multiple single disease care plans. To integrate care, allows the agents to become aware of new risks. Risks they are unaware of when they are blind to each others activities. Integrated care is a requirement for pro-active care. When care is integrated, the new risks that are identified can be managed, so that risks can be eliminated or lowered.

Safe care or pro-active care is the active identification of risks and management of those risks. It is useful for any patient in an unstable condition, where deterioration might be heralded by subtle changes that are hard to detect. Risk management arises from actively recognizing risks, monitoring risks, and responding to them with action.

Person-centered care is the co-creation of a care plan, together with the patient. It is a sharing of power, so that what is known to be effective and meaningful by the patient, is reflected in the care plan. PCC is also a pre-requisite for safe care. Unless the patient finds the risk-mitigation activities both meaningful and feasible, he/she will not implement them. Patients can veto anything we suggest. Therefore, involving and engaging patients are part and parcel of safe care.

I would invite the authors to re-review their introduction in light of a conceptual distinction between integrated care and safe care, while recognizing their interdependencies. Safety does not arise from integration itself. It arises from the improved opportunity for risk management.

We are grateful to the reviewer for these comments and agree some clarity in terms of conceptual distinction is required. We have revised the introduction in light of these comments and explained how integration alone may not assure safer care but has an important role to play in enabling more effective risk management – see line 67-88. We have also further acknowledged this aspect in the conclusion section.

Table 1 is rather confusing.

It seems to me that table 1 descriptions of integrated care conflicts with the excellent descriptions given in Box 1. Also, the patient safety descriptions in Table 1 are not very meaningful to illustrate how care becomes safer as a result of care integration. I suggest that integrated care by level of system is described either in introduction or in methods, and that you retain the descriptions in Box 1.

I suggest you consider the following: Safe care arises from integrated care at micro-level:” ... by addressing the risks posed by multiple referrals, handovers and discharges that arise as the responsibility for care crosses care boundaries and involves different practitioners, teams, organisations and systems” AND identifying concrete risks which are then managed by the providers working together.

I believe both macro and meso-level integration of services lead to safer care, mainly by supporting a better micro-level opportunity to identify and manage risks. Hence I think table 1 is superfluous.

We have removed table 1 from the manuscript.

Safety in the community setting:

The authors write:

“Even so, integrated care initiatives themselves pose a risk to patient safety as care shifts away from hospitals and closer to home. Adverse events in the community setting cause between 8-12% of hospital admissions, around a half of which are thought to be preventable. Harm in this setting may result from medication mismanagement, poor infection control, failure to ensure safe mobilisation or a lack of prevention of and care for pressure sores/ulcers.” To me, this paragraph tells me that patients with complex care needs are safer in hospital than at home.

However, we know that hospitalization increases the risk of hospital acquired infections and deconditioning. {Smith, 2020 #14379} Being hospitalized is also not a good quality of life. Please revise.

This paragraph has been revised with no reference to hospital care, see line 89-90.

Search terms:

The search terms are outlined at a very high level in the methods section. Please outline the key conceptual words that would define the areas that you have combined in your search.

We now include a few overarching search terms. See line 139-141.

Single or multiple settings?

In methods you state that “inclusion criteria: Involved at least two different types of organisations e.g. hospital”. Yet in results you report on several papers set in a “single setting”. Then again in results, you explain what you mean with the single setting: “Studies based in a single setting included approaches that involved professionals from different sectors across social, acute, primary and community working together (often in the same team) to provide care.” This is confusing. Please revise and use a term that is intuitive to describe a single team, which consists of professionals representing many settings. Maybe multidisciplinary team ?

We thank the reviewer for this comment. We have revised this accordingly. See the updated table 1 and line 208. We hope our reference to ‘groups or teams of professionals from different sectors’ provides more clarity.

Measures for assessing impact:

The authors have chosen, appropriately, to use Donabedians Structure > Process > Outcomes to organize their findings. However you report only on process and outcomes measures of impact. Why did you leave out structural measures of impact? Example. The implementation of a policy to increase the number of multi-disciplinary teams (MDT), would be expected to increase the number of MDTs (structures) that can handle integrated care.

We omitted structural measures as none of the studies included measures that could be categorised as structural – for example, there were no studies referring to implementation of a policy such as increasing the number of MDTs or the implementation of system wide patient electronic records. The studies involving integrated care records were specifically focussed on outcome measurement e.g. safety incidents not on the implementation of the records and their extent. We acknowledge the lack of structural measures as a potential limitation.

Self-care and risk management

One of the most important ways of both monitoring and reducing risk is to involve and engage patients in this activity. Patients are the only persons present at all times in their own patient journey. They are the ones who have most to gain by effective risk management. They are also much cheaper labor and are always present. It is almost never possible to promote effective risk management without a proper involvement of patients.

As the authors did not include self-care as a risk-management focus in their search, I suggest that this is mentioned in the limitations of the study.

We have now referred to this as a limitation in the appropriate section of the manuscript. See line 369.

January 2023 responses

General summary

Integrated care is thought to promote safer care, through improved coordination and quality of care. This is a scoping review, seeking to tease out the safety process and outcomes measures that arise as a benefit from integrated care interventions. They seek to understand what integrated care interventions that also report on patient safety look like (context, and content), which RQ they answer, and especially, which patient safety issues are addressed, including safety outcomes.

The authors have performed a broad search, including the grey literature. They present the identification of relevant studies in a clear and comprehensive manner, including a flow-chart. The authors have applied clear inclusion/ exclusion criteria, They use clear definitions of integrated care (across at least 2 organizations) and patient safety as defined by WHO. They have structured their review by micro- meso and macro-level interventions.

The authors included 24 studies. They find that integrated care and safety interventions are mostly focused on medication review and falls preventions. The interventions vary widely, from macro: system-wide implementations of integrated care EHRs, meso: development of care pathways, to micro: person-centered care plans. They conclude that there is a lack of focus on safety specifically in integrated care interventions.

I congratulate the authors with a big task, which surely has taken a lot of time and resources. Thank you. The study is methodologically stringent, and the analysis is clear, the language is crisp and clear.

Major points

Unclear research question.

Are you studying the safety effects of integrated care, or are you studying risk management strategies that are often employed together with integrated care approaches?

I agree with the authors, that integrated care could impact safety, as continuity of care, could secondarily impact safety. The authors describe how fragmented care carries an inherent risks for medication errors, pressure sores, infections, unsafe mobilization etc. The authors offer the following theoretical understanding of why integrated care should be safer:

“Risks to patient safety can be mitigated within ICS by strengthening some of the key structural and relational aspects of service provision such as communication, partnership working, care coordination and person-centred care as well as intervening to reduce specific harms.”

In my opinion, person-centered care, and specific risk reduction strategies are not in and of themselves inherent to or part of integrated care. The challenge is to differentiate between risk-management that arises from integrated care, and risk-management approaches that are independent of, and therefore not really a result of integrated care. For instance, a falls prevention program can be done by a physiotherapist alone, or it could be performed by a physio and the pharmacist together. When they combine their competencies, they will create a more comprehensive

falls prevention plan than each could do on their own. The latter would be a patient safety improvement which can be attributed to integrated care, whereas the former is not attributable to integrated care. The former would be a risk-management strategy, made in a stand-alone context, by the physiotherapist.

Another way to think about integrated care and safety is that information always precedes action. The integrated care team may be safer because the care team is sharing information and competencies. Therefore, they can understand and manage risks better, when they have a broader overview of the case, than each of the agents can, when they work on their own.

To summarize, I believe it is essential that we learn to distinguish between different strategies of care quality improvement. To reach the quadruple aim, we need to pursue person-centered, integrated and pro-active care or patient safety, simultaneously {Berntsen, 2019 #12330}. We need to understand the essence of each term, and not lump them all together under one heading. Thus the paper would be strengthened by a theoretical reflection on how the authors would expect integrated care to improve safety. Secondly to differentiate between patient safety improvements that arise from integrated care, and those that arise from improved risk-management, but are not really dependent on the integrated care component. I would invite the authors to re-think their research question in terms of these issues.

The reviewer provides some very useful insights. That said, as the aim of the review mentions our primary interest was to understand how integrated care approaches could improve patient safety. We refer to this throughout, including in the discussion sections when we discuss how integrated care at the different levels of the system may improve patient safety by minimising risk. We agree that risk reduction strategies are not inherently part of integrated care approaches – but our study highlights how integrated care could reduce risk across a system. We also believe the person-centred approaches are now a key underlying principle of integrated care approaches, although whether this is adequately delivered upon is an entirely different matter, outside the scope of this study.

Search strategy

The terminology used for patient safety in the search strategy is quite narrow. I miss a comment on the plethora of terms other than patient safety, that could have been part of the search strategy. These are terms which also indicate an active management of risks, such as pro-active care, anticipatory care, risk-management, remote follow up, early intervention, self-mangement of risk, etc.

While we recognise that there was scope to expand the list of terms associated with patient safety, in our consultation with the wider research group (including safety experts) the chosen terms were deemed to adequately align with the definition of patient safety.

Single organization contexts – can they be integrated?

In Line167. Please explain how papers set in single organization interventions, such as either hospital or community health centers, can call themselves integrated?

This has been clarified– line 194-196. Despite the studies being set in a single setting, the interventions delivered involved different professionals across different sectors i.e. acute, primary, social care etc.

Why are doctors not part of integrated care interventions?

In line 243: "... but doctors were not involved.." this is very surprising. Were there no doctors involved in any of the integrated care approaches? If this is so, I believe it warrants some reflection in the discussion. Why are doctors not part of the integrated care effort?

Thank you for highlighting this issue. We have included some brief reflections in the discussion see lines 333-338.

What role should patients have in patient safety efforts?

Line358: “None of the studies in this review collected data on the experience of patients and carers alongside clinical outcomes” I agree that this is an important point. Could this be the result of the way you conducted your search?, i.e. you searched for integrated care, not person-centered care, and thus inadvertently excluded studies that were focused on both topics? If yes – add a sentence on this in weaknesses and limitation.

Thank you – this has been addressed, lines 355-358.

Minor comments:

Page 5, line 70: “... to ensure safe mobilisation or a lack of prevention of and care for pressure areas. 12”

What kind of pressure are we talking about here? Pressure sores? Or pressure in work for professionals?

Acronyms must be spelled out. MDT, ICS, ADE – please provide a full-name for each, before you apply the abbreviation.

Minor comments have been addressed as advised.

Thank you for the opportunity to review the manuscript: “Approaches to improving patient safety in integrated care: a scoping review” for BMJ Open. The article used a scoping review approach to establish the approaches employed to manage patient safety in integrated care for community dwelling people with long term conditions.

This was a solidly designed piece of work that followed procedures of the increasingly popular scoping review methodology. Apart from some minor editorial points there were a couple of fundamental issues that need to be addressed in any review of the manuscript.

There are three areas of concern that I feel would be worth close consideration.

The theory and the logic of the study

The authors have set themselves a difficult task in this work. The concepts of integration and safety are both complex, however I am concerned that there has been very little use of theoretical considerations to inform the review. – I feel that the review would have benefited considerably to have incorporated a broader conceptualisation of integration – it would have been helpful to refer to Valentijn, P. P., et al. (2013). "Understanding integrated care: a comprehensive conceptual

framework based on the integrative functions of primary care." *Int J Integr Care* 13: e010. In terms of safety, again there seems no real conceptual model – again it would have been worth at least acknowledging models such as the Manchester Patient Safety Framework would have been helpful.

We thank the reviewer for their comments. We have addressed this issue by summarising the relevant aspects of Valentijn et al's model and added further detail from the OECD safety governance framework. See table 1.

The population of focus.

Neither the title, the research question nor the abstract characterises a key limitation of the review – the inclusion only of patient populations of individuals aged 65 and above. As with the intervention / approaches issue, the reader of the article would be expecting not to see the restriction. It would be especially irritating for clinicians and consumers aware of the challenges faced by younger individuals living with long term disabilities, especially those requiring long term integrated care.

We grateful to the reviewer for highlighting this issue. The review does in fact include four studies among the included list of studies in which the population of adults is over 18. We have therefore amended the inclusion criteria accordingly. The search terms include 'vulnerable adults' which would have accounted for those adults with long term conditions that are not necessarily over 65. We believe the framing of the inclusion criteria as restricting to over 65 was an oversight on our part.

A blurring of approaches and interventions.

The title and the stated objective of the scoping review is to identify approaches that have been used to manage patient safety in integrated care. By contrast, the review seems devoted to identifying interventions. The confusion is compounded by the fact that the background speaks of the varied levels of systems of care, and of the structures that have been used to address and incorporate safety. Then, at line 79 on page 4 the language transitions to speaking of interventions.

The remainder of the article then speaks about interventions. Consistency would help the reader.

Otherwise I have made a few comments about the methods and some editorial issues

We thank the reviewer for pointing this out. We have used approaches throughout to reduce any ambiguity.

Methods

The paper seems to follow the Arksey and O'Malley approach (although I would suggest that the authors reword the phrase 'We used the first five stages of Arksey and O'Malley's framework...' (p4 line 89) – it gives the impression that a phase was left out. The 'left out phase is optional in the framework.

This has been rectified, thank you.

As far as the methods were concerned, there is a phrase that justified why a scoping review is the most appropriate review in the methods (p5 line 91) – it would be better in the background as the justification. It is not entirely clear as to how the authors have a) identified the gap in understanding and b) decided that a scoping review is the way to fill the gap. Or, as is said in PRISMA-Scr “Describe the rationale for the review in the context of what is already known. Explain why the review questions/objectives lend themselves to a scoping review approach”

We have heeded the advice of the reviewer moving the justification sentence to the background section and adding more clarity to the rationale for the study. See lines 87-95.

Finally, I was surprised that only one reviewer screened titles and abstracts – I couldn't be clear as to how 'borderline' articles were considered by the research team.

Many thanks for this comment – all studies screened by the first reviewer for which there was uncertainty about eligibility criteria were in fact subject to a further check by the second reviewer. This has been stated in line 132.

Editorial issues.

Acronyms can occasionally be helpful in improving flow of a document. However the authors have used several acronyms that, if anything make it harder to understand the article. The general reader would be familiar with 'QI' for example, but would have difficulty with LTC ICS ADE, and MDT – they make clarity more difficult. The use of LTC is even more problematic given the acronym usually refers to long term care (as opposed to long term conditions)

The written English is otherwise clear, although there are several minor typographical errors through the manuscript (mainly missing prepositions).

Acronyms have been addressed as follows:

LTC and ICS has been spelled fully in all cases. We regard ADE and MDT as fairly commonly used and understood acronyms both by practitioners and in the literature.

Thank you for the opportunity to review the manuscript. Overall, this is a well written manuscript, however, I have the following comments for the authors to consider.

1. There are inconsistencies in the alignment of the research question, the results and the conclusion. The focus of the scoping review is to map the approaches employed to manage patient safety in integrated care for community dwelling people with long term conditions. The conclusion of the abstract includes evaluation, the conclusion of the manuscript introduces new information and does not summarise the results of the review.

We have removed reference to evaluation in the conclusion of the abstract. The conclusion of the paper now aligns with the abstract but also summarises the paper's main findings i.e. the various approaches associated with integrated care that improve patient safety.

2. The concept of the review should be "integrated care approaches for chronic conditions. The authors only rereferred to two or more interventions (please provide a reference for this). these two phrases are not similar.

We thank the reviewer for this comment. To ensure consistency we have removed reference to intervention and have used approaches throughout.

3. Please provide a definition for "integrated care approaches in the introduction.

We have expanded the definition of integrated care in the introduction including adding further relevant content in table 1.

4. I am not sure if the focus is patient safety or system safety, please use consistent terms.

We now refer largely to patient safety other than when discussing outcome indicators for safety systems.

5. scoping reviews protocols should be published in open science framework. Please refer to this paper Peters, M.D., Marnie, C., Tricco, A.C., Pollock, D., Munn, Z., Alexander, L., McInerney, P., Godfrey, C.M. and Khalil, H., 2020. Updated methodological guidance for the conduct of scoping reviews. JBI evidence synthesis, 18(10), pp.2119-2126.

We thank the reviewer for this comment but there is not mandatory requirement to publish a scoping review protocol and we cannot see the value in doing so retrospectively?

6. Please try and present some of the data in graphical or tabular format. It is hard to include all the results in one table. We did consider splitting up parts of the overall table but found that having all the results in one table was the most cogent approach to presenting the results.

7. Please provide other examples of patient safety targets in your discussion.

We thank the reviewer for this comment and have added three further examples of patient safety targets. See line 421-423.

VERSION 3 – REVIEW

REVIEWER	Berntsen, Gro Norwegian centre for E-health Research, Integrated care
REVIEW RETURNED	13-Mar-2023
GENERAL COMMENTS	The authors have responded adequately to my previous comments. No further comments this time. Congratulations with a well performed scoping review on an important topic.

VERSION 3 – AUTHOR RESPONSE